# LLMs Can Generate a Better Answer by Aggregating Their Own Responses

## Abstract

Large Language Models (LLMs) have shown remarkable capabilities across tasks, yet they often require additional prompting techniques when facing complex problems. While approaches like self-correction and response selection have emerged as popular solutions, recent studies have shown these methods perform poorly when relying on the LLM itself to provide feedback or selection criteria. We argue this limitation stems from the fact that common LLM post-training procedures lack explicit supervision for discriminative judgment tasks. In this paper, we propose Generative Self-Aggregation (GSA), a simple and effective prompting method that improves answer quality without requiring the model's discriminative capabilities. GSA first samples multiple diverse responses from the LLM, then aggregates them to obtain an improved solution. Unlike previous approaches, our method does not require the LLM to correct errors or compare response quality; instead, it leverages the model's generative abilities to synthesize a new response based on the context of multiple samples. While GSA shares similarities with the self-consistency (SC) approach for response aggregation, SC requires specific verifiable tokens to enable majority voting. In contrast, our approach is more general and can be applied to open-ended tasks. Empirical evaluation demonstrates that GSA effectively improves response quality across various tasks, including mathematical reasoning, knowledge-based problems, and open-ended generation tasks such as code synthesis and conversational responses.

## 1 Introduction

Large Language Models (LLMs) have demonstrated remarkable capabilities across a wide range of tasks, yet they often struggle with complex problems requiring careful deliberation or multi-step reasoning (Bang et al., 2023). This limitation has prompted the development of various prompting techniques, with self-correction (Kamoi et al., 2024; Madaan et al., 2023) and choose-from-N (Bai et al., 2022; Snell et al., 2024; Chen et al., 2023) emerging as popular approaches. In self-correction, models revise their initial responses based on feedback, while choose-from-N methods generate multiple candidates and select the best one. However, recent studies have shown that these methods perform suboptimally when relying on the LLM itself to provide feedback or selection criteria without external guidance (Huang et al., 2024; Mahan et al., 2024).

We argue that this phenomenon reflects an asymmetry in standard LLM training: while next-token prediction enables strong zero-shot *generative* ability, it provides little explicit supervision for *discriminative* judgment (e.g., scoring, pairwise comparison). Recent work on generative reward models Mahan et al. (2024) supports this view by showing that specialized training can substantially enhance judgment capabilities, albeit at the cost of additional data curation and compute that may be impractical for many applications.

To the best of our knowledge, existing methods that successfully self-improve without additional training on discriminative processes fall into two main categories: approaches utilizing external feedback (e.g., code executors (Stengel-Eskin et al., 2024) or computational verifiers (Pan et al., 2023)) and the self-consistency method (Wang et al., 2023). The external feedback approach requires careful task-specific design and is not universally applicable across different domains. Self-consistency takes a different approach by generating multiple samples and employing majority voting among them. However, its applicability is limited to tasks with specific, verifiable tokens as answers, such

as mathematical reasoning or multiple-choice questions, making it unsuitable for open-ended tasks. Moreover, self-consistency can only aggregate final answers, overlooking the valuable reasoning processes behind these answers. By contrast, selection-based variants that ask an LLM to pick among candidates (e.g., Universal self-consistency (Chen et al., 2023)) still require reliable *discriminative* judgments from the model itself, which prior studies suggest can be fragile without external signals Huang et al. (2024); Mahan et al. (2024).

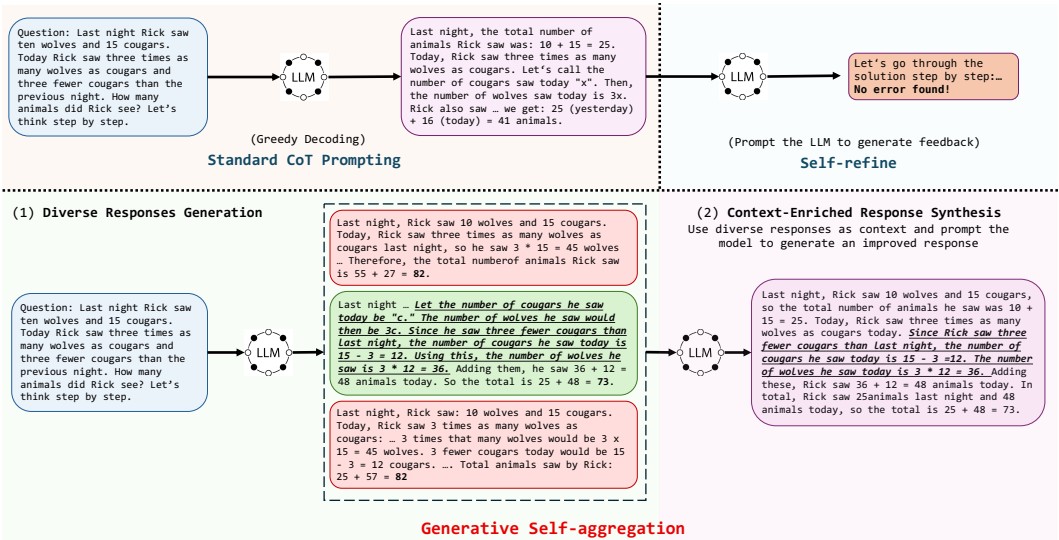

Figure 1: Illustration of Generative Self-aggregation with an example on a math problem using Llama 3 8B as the language model.

Building on these insights, we propose Generative Self-Aggregation (GSA), which improves answer quality by generatively aggregating information from multiple responses. Unlike previous methods, GSA does not require the model to explicitly judge or compare responses and does not need additional training. As illustrated in Figure 1, GSA first generates multiple diverse responses, then uses these responses as context to prompt the model to aggregate them and generate an improved solution. The common LLMs' training paradigm of predicting subsequent tokens based on input context allows the model to identify and learn from stronger solutions through its natural text generation capabilities, enabling the model to combine strengths from different solutions. Unlike traditional self-consistency which relies on majority voting, our approach leverages the generative power of LLM for aggregating multiple responses. By utilizing the reasoning process rather than just the final answers, the model has access to more information that can be aggregated to improve the final solution. Moreover, this generative aggregation approach extends beyond specific-answer tasks to open-ended problems where majority voting would not be applicable.

Through extensive experimentation across mathematical reasoning, knowledge-based, and open-ended generation tasks, we demonstrate that our method outperforms both self-correction and choose-from-N baselines across different tasks and model scales. Our method achieves comparable or better performance to self-consistency on tasks with verifiable answers, while self-consistency cannot be applied to open-ended tasks. Our ablation studies demonstrate the method's robustness across different sampling strategies. Further analysis of likelihood distributions reveals that LLMs are more confident on generating new responses than selecting among existing ones, providing empirical support for our framework.

The rest of the paper is organized as follows: Section 2 reviews related work in LLM prompting techniques; Section 3 introduces our proposed Generative Self-Aggregation approach; Section 4 presents comprehensive experimental results; Section 5 draws a brief conclusion and discusses potential future directions.

## 2 RELATED WORK

### 2.1 SELF-CONSISTENCY

Self-consistency (Wang et al., 2023) is a decoding strategy that improves Chain-of-Thought (CoT) prompting by leveraging multiple reasoning paths. Instead of using greedy decoding to generate a single solution, it first samples multiple diverse reasoning paths and then aggregates their final answers through majority voting to determine the most consistent one. Its success on mathematical reasoning and common sense questions demonstrates LLMs' ability to generate correct solutions across different attempts. Lin et al. (2024) further extends this concept and employ LLM to extract answer and conduct voting on various reasoning paths. However, their application to open-ended tasks remains challenging due to the lack of clear voting mechanisms.

### 2.2 SELF-CORRECTION

Self-correction in Large Language Models has emerged as a significant research direction, focusing on models' ability to recognize and improve their outputs based on feedback (Kamoi et al., 2024; Paul et al., 2024; Madaan et al., 2023). A substantial body of work explores self-correction with external feedback sources, such as human annotations (Shinn et al., 2023), code executors (Stengel-Eskin et al., 2024; Chen et al., 2024b; Gou et al., 2024), or symbolic reasoning tools (Pan et al., 2023). While effective, these approaches are limited by their reliance on additional knowledge sources or tools that may not always be available. When only the language model itself is available, researchers have proposed various intrinsic self-correction methods. Self-Refine (Madaan et al., 2023) and RIC prompting (Kim et al., 2023) prompt the model to provide feedback on and refine its previous outputs. However, recent studies have shown that these approaches, which rely on LLMs' ability to make discriminative judgments about their own outputs, often produce inaccurate assessments and yield suboptimal results (Huang et al., 2024).

To address these limitations, some researchers have explored training-based approaches to enhance self-correction capabilities. These include supervised fine-tuning methods (Paul et al., 2024; Welleck et al., 2023) and reinforcement learning approaches (Akyürek et al., 2023). While promising, these methods require substantial human-annotated training data, limiting their practical applicability.

### 2.3 CHOOSE-FROM-N METHODS

Another prominent approach to improve language model outputs is the choose-from-N paradigm, where multiple candidate responses are first generated and then selected based on specific selection criteria. Best-of-N sampling (Stiennon et al., 2020; Jinnai et al., 2024) represents a widely adopted variant of this approach, utilizing reward model scores as selection criteria during decoding to better align responses with human preferences. This methodology has been extended by works incorporating specialized verifiers or process reward models (PRM, Snell et al. (2024)) to enhance selection accuracy. However, the effectiveness of these approaches heavily depends on having well-trained reward models that accurately reflect human preferences.

An alternative strategy employs the language model itself as the evaluators for selection. For instance, Constitutional AI (Bai et al., 2022) introduces Reinforcement Learning from AI Feedback (RLAIF), employing LLMs to identify harmful content and generate preference labels for training. Universal self-consistency (Chen et al., 2023) employs LLMs to select the best answer from several candidates. Zheng et al. (2023) further establish that strong LLMs can provide judgments that correlates with human preferences. While this correlation can be effective for tasks like model evaluation or dataset construction where errors can be averaged out, using LLMs as judges for selecting better responses in individual cases may not be optimal. Moreover, recent studies (Fu et al., 2023; Thakur et al., 2024) have identified several challenges in these approaches. Mahan et al. (2024) shows that LLMs' zero-shot judgments may not always fully align with human preferences. Thakur et al. (2024) demonstrates that recent open-source LLMs' alignment performance falls considerably short of human-to-human agreement, with their evaluations often deviating significantly from human assessments. Our method takes an alternative approach without requiring LLMs' discriminative judging capabilities. (Farinhas et al., 2023) focus on translation tasks and propose two variants: choose or generate based on previous translations, which bears some similarity to our work. However, their methods show little improvement when using the LLM itself, requiring external guidance.

### 2.4 MULTI-MODEL COLLABORATION

Recent research has explored the potential of leveraging multiple large language models as interactive agents to solve complex problems. Various multi-agent frameworks have emerged where models assume distinct roles, such as debaters and judges (Du et al., 2024; Wang et al., 2024b). For example, Chen et al. (2024a) present an iterative discussion framework that enables multiple LLM agents to engage in round-table discussions with confidence-weighted voting, where it require models to provide a confidence score. Wang et al. (2024a) propose a multi-layer design to iteratively aggregate responses from different models. However, their fundamental focus differs from ours - they aim to leverage the complementary strengths across heterogeneous models, while our method explores how a single model can improve over its own responses. Moreover, all these multi-agent approaches require simultaneous deployment of multiple models and involve substantially more complex frameworks, resulting in substantially higher computational and resource costs. In contrast, our work introduces a lightweight, single-round, single-model method that avoids explicit discriminative judgments and achieves strong empirical performance across diverse tasks.

## 3 METHODOLOGY

We propose Generative Self-Aggregation (GSA), a prompting method that improves answer quality without relying on LLMs' discriminative ability. Our method consists of two key steps: (1) diverse response generation and (2) context-enriched response synthesis. Unlike traditional self-consistency, which relies on majority voting to find agreement among multiple final outputs, our approach enables the model to synthesize an improved solution by learning from diverse attempts. Our method operates within the model's generative framework without requiring any discriminative judgments (such as selecting or judging responses) and does not require additional training.

### 3.1 DIVERSE RESPONSE GENERATION

Given a language model $\mathcal{M}(\cdot)$ and query $q$, we first generate $n$ diverse responses by sampling from the model's distribution:

$$r_i \sim \mathcal{M}(r|q), \quad i = 1, \ldots, n.$$

We can employ various sampling strategies, such as temperature sampling (Ackley et al., 1985) or nucleus sampling (Holtzman et al., 2020), to generate these candidates. The diversity of these responses is crucial for providing rich context in the subsequent synthesis step. For example, in mathematical reasoning tasks, diverse candidates may explore different solution paths, potentially containing valuable correct intermediate steps even when reaching incorrect final answers. In knowledge-based tasks, they can access different aspects of the model's internal knowledge that a single deterministic generation might miss.

### 3.2 CONTEXT-ENRICHED RESPONSE SYNTHESIS

After generating diverse candidates, we construct an enriched prompt by combining the original task query with the generated responses and ask the model to generate an new response:

$$r' \sim \mathcal{M}(r|\text{Prompt}(q, \{r_i\}_{i=1,\ldots,n}).$$

Standard training of next-token prediction enables the language model to attend to and learn from useful content in the provided context. By providing the model multiple attempts, we enable it to identify effective reasoning patterns or knowledge and combine them into a more refined response, potentially combining the strengths while avoiding their individual weaknesses. The following box presents a prompt template that we use for a coding task MBPP (Austin et al., 2021):

```
### Here is the problem:
{query}
### Reference Solutions:
{diverse_responses}
### Instructions:
1. Review the above solutions.
```

```
2. Generate a Python function that solves the Problem.
3. Provide a brief explanation of your reasoning.
4. Ensure your code is enclosed within a ```python``` code block.
```

Figure 1 presents an example using Llama 3 8B (Dubey et al., 2024) on a mathematical reasoning task. When given this problem, standard zero-shot chain-of-thought prompting with greedy decoding produces an incorrect solution. Self-refine ask the model to provide a feedback to the generated solution, but failing to correctly identify the errors. Our method, in contrast, first generates three diverse solutions through sampling, with two arriving at an incorrect answer and one reaching the correct answer. Instead of attempting to select the final answer through voting, our method provides these diverse attempts as context and prompts the model to generate an improved solution.

The resulting response not only arrives at the correct answer but also demonstrates an evolution in reasoning strategy: the original correct solution first derives the number of wolves as $3c$ with $c$ defined as the number of cougars, and then calculate $c$, following the sequential order of conditions in the question. Differently, the improved solution takes a more direct approach without introducing variables. This transformation suggests that the model can not only identify correct reasoning patterns from the provided attempts but also simplify them into more straightforward solution paths.

## 4 EXPERIMENTS

### 4.1 EXPERIMENTAL SETUP

**Language models.** We evaluate our method using four different LLMs to ensure robustness across model scales and architectures. We use the instruction-tuned version of the open-sourced models for experiments.

We evaluate on four instruction-tuned models: **Llama 3 8B** (Dubey et al., 2024), trained on 15T tokens; **Gemma 2 9B** (Rivière et al., 2024), a distilled 9B model with architectural improvements; **Qwen 2.5 14B** (Yang et al., 2024), which improves over earlier versions in reasoning and knowledge; and **GPT-4o Mini**, a compact GPT-4 variant optimized for speed and cost with strong generation quality.

**Tasks and datasets.** We conduct comprehensive evaluations across diverse tasks with task-specific evaluation metrics:

• **Mathematics reasoning.** For mathematical reasoning capabilities, we use Grade School Math 8K (GSM8K; Cobbe et al. 2021), a dataset of grade school math word problems that require multi-step reasoning. We also include Math (Hendrycks et al., 2021b), a diverse collection of mathematical problems spanning algebra, arithmetic, and geometry. We follow the original paper to use string match for accuracy calculation. SVAMP (Patel et al., 2021) serves as a challenge set designed for more robust evaluation of models against elementary level math word problems.

• **Knowledge Tasks.** For testing knowledge application, we employ the Massive Multitask Language Understanding benchmark (MMLU; Hendrycks et al. 2021a), which covers 57 subjects ranging from mathematics to law. Due to resource constraints, we randomly sample 10% of the MMLU test set for evaluation. Additionally, we use Graduate-Level Google-Proof Q&A Benchmark (GPQA; Rein et al. 2023), a benchmark with very hard question written by experts in biology, physics, and chemistry, which also requires strong reasoning ability.

• **Open-ended Tasks.** To evaluate performance on less constrained tasks, we use MT-bench (Zheng et al., 2023), a multi-turn dialogue benchmark specifically designed to assess open-ended conversation capabilities. Alpaca Eval (Li et al., 2023) provides a comprehensive suite for evaluating instruction-following abilities. For programming tasks, we include MBPP (Austin et al., 2021), which contains 974 basic Python programming problems. For MT-bench, we employ GPT-4 (OpenAI, 2023) as the evaluator to score the response. For Alpaca eval, we employ GPT-4 to calculate win rate against responses from GPT4-turbo.

**Baselines.** We compare against five baselines: **Greedy**, which uses standard decoding with temperature = 0; **Self-Refine** (Madaan et al., 2023), which iteratively improves responses based on model-generated feedback; **Self-Consistency** (Wang et al., 2023), which samples multiple outputs

and selects the most frequent answer via majority vote; **Universal Self-Consistency** (Chen et al., 2023), which generates multiple responses and selects one based on a model-predicted index; and **Best-of-N (Oracle)**, an upper bound that marks a sample as correct if any candidate is correct (for MT-bench, we report the best score among responses).

To ensure fair comparison, we standardize computational budget across all methods by fixing the number of model calls to 4 in the main results. We set this limit for practical cost considerations, as the cost scales linearly with the number of class and the largest performance gains typically occur within the first few responses. Nevertheless, we experiment our method with larger number of responses in Section 4.3.

For Self-Consistency, we generate four diverse candidates using temperature sampling and apply majority voting on the final answer. We randomly select three of these candidates for our method and universal self-consistency. For Self-Refine, we limit the feedback-refinement loop to 2 iterations. We maintain consistent prompt templates across our method and universal self-consistency, while adhering to the original prompts for Self-Refine. In the final aggregation step of our method, we employ greedy decoding for closed-ended tasks to ensure deterministic outputs, and temperature sampling for open-ended tasks, following standard practice. For additional case study, detailed prompts and parameter settings, please refer to the Appendix.

## 4.2 MAIN RESULTS

Table 1: Performance comparison on Llama 3 and GPT 4o mini across tasks, and methods. GSM8K, Math, GPQA, MMLU, SVAMP and MBPP scores are accuracy percentages. MT-bench is on a 1-10 scale; Alpaca scores are length-control win rate percentages against responses from GPT4-turbo; Best results for each model-task pair are in **bold**.

| Model | Method | Tasks | | | | | | | |
|---|---|---|---|---|---|---|---|---|---|
| | | GSM8K | MATH | GPQA | MMLU | SVAMP | MT-bench | Alpaca | MBPP |
| Llama 3 | Best-of-N (Oracle) | 91.74 | 47.26 | 61.23 | 77.08 | 92.67 | 8.04 | 36.14 | 60.60 |
| | Greedy | 82.47 | 29.28 | 32.14 | 63.13 | 82.67 | 7.43 | 27.55 | **55.20** |
| | Self-Refine | 82.99 | 30.32 | 32.14 | 63.53 | 83.00 | 7.30 | 24.42 | 52.80 |
| | Self-Consistency | **86.35** | 31.68 | 33.26 | **65.62** | **88.33** | N/A | N/A | N/A |
| | Uni. Self-Consistency | 84.99 | 31.28 | 33.26 | 64.48 | 87.33 | 7.45 | 29.14 | 53.20 |
| | Ours | 86.05 | **32.46** | **35.04** | **65.62** | 88.00 | **7.53** | **29.34** | **55.20** |
| GPT 4o Mini | Best-of-N (Oracle) | 96.29 | 86.72 | 56.47 | 86.25 | 95.33 | 9.30 | 61.67 | 78.60 |
| | Greedy | 93.48 | 76.54 | 41.07 | 80.85 | 93.00 | 8.95 | 47.99 | 73.20 |
| | Self-Refine | 92.42 | 76.44 | 40.03 | 83.03 | 91.00 | 9.02 | 51.60 | 70.60 |
| | Self-Consistency | **94.77** | 77.26 | 40.40 | 85.00 | 93.00 | N/A | N/A | N/A |
| | Uni. Self-Consistency | 94.47 | 76.80 | 38.39 | 80.51 | **94.00** | 8.84 | 50.20 | 72.60 |
| | Ours | 94.69 | **78.25** | **42.17** | **85.11** | **94.00** | **9.13** | **55.85** | **74.20** |

Table 2: Performance comparison on Gemma 2 and Qwen2.5 across tasks. GSM8K, MATH, GPQA, MMLU, SVAMP, and MBPP scores are accuracy percentages. MATH is evaluated on a 500 sample subsets and Alpaca-eval is excluded due to resource constraints. MT-bench is on a 1-10 scale.

| Model | Method | Tasks | | | | | | |
|---|---|---|---|---|---|---|---|---|
| | | GSM8K | MATH | GPQA | MMLU | SVAMP | MT-bench | MBPP |
| Gemma 2 9B | Best-of-N (Oracle) | 93.03 | 84.00 | 50.45 | 78.43 | 91.33 | 8.74 | 61.80 |
| | Greedy | 87.11 | 73.20 | 31.74 | 65.91 | 88.00 | 7.99 | **57.60** |
| | Self-Refine | 87.04 | 72.60 | 31.70 | 65.43 | 87.67 | **8.35** | 57.40 |
| | Self-Consistency | 89.08 | 75.00 | 33.26 | 69.04 | 88.33 | N/A | N/A |
| | Uni. Self-Consistency | 88.25 | 72.80 | 31.92 | 65.84 | 86.33 | 8.30 | 57.20 |
| | GSA (Ours) | **89.61** | **76.00** | **34.82** | **69.11** | **88.33** | **8.35** | 57.40 |
| Qwen 2.5 14B | Best-of-N (Oracle) | 97.19 | 98.00 | 63.17 | 86.41 | 95.00 | 9.22 | 80.20 |
| | Greedy | 94.62 | 94.40 | 39.51 | 79.07 | 92.33 | 8.63 | 72.00 |
| | Self-Refine | 94.62 | 94.60 | 40.18 | 79.34 | 92.67 | 8.65 | 71.80 |
| | Self-Consistency | 95.68 | 95.80 | 40.40 | 79.79 | 93.00 | N/A | N/A |
| | Uni. Self-Consistency | 95.68 | 95.40 | 40.62 | 79.64 | 92.67 | 8.66 | 73.00 |
| | GSA (Ours) | **95.75** | **96.20** | **41.74** | **79.92** | **93.00** | **8.99** | **73.80** |

Table 1 summarizes performance across models, tasks, and methods. Our approach consistently outperforms both universal self-consistency and self-refine, demonstrating the advantage of generative aggregation over model-based selection or feedback. Notably, universal self-consistency and self-refine sometimes underperform compared to greedy decoding (e.g., GPQA with GPT-4o Mini), supporting our hypothesis that standard LLM training does not sufficiently develop discriminative capabilities.

The Best-of-N (Oracle) results show varying headroom for improvement, with larger gains from our method when oracle accuracy significantly exceeds greedy decoding—indicating GSA effectively exploits diverse, high-quality candidates.

On mathematical and knowledge-based tasks, our method matches or surpasses self-consistency despite using fewer candidates (3 vs. 4). With Llama 3, we achieve comparable performance on GSM8K and MMLU and improvements on MATH and GPQA. For open-ended tasks where self-consistency is inapplicable, GSA still yields consistent gains. Notably, with GPT-4o Mini, we observe strong improvements on Alpaca-Eval and MT-bench, highlighting the method's broad applicability across both structured and open-ended settings.

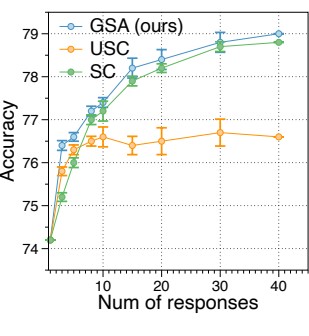

Figure 2: Comparison with different number of responses on MATH with GPT 4o mini.

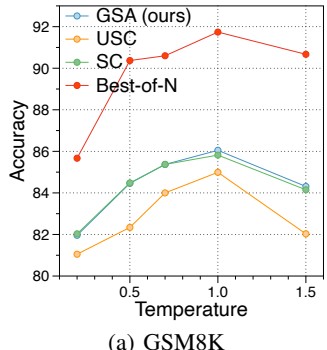
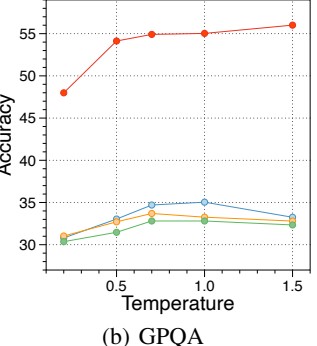

(a) GSM8K      (b) GPQA

Figure 3: Comparison of baselines performance with different temperature and $N = 3$ on GSM8K and GPQA with Llama 3 8B.

### 4.3 Ablation Studies

**Number of responses.** We analyze how the number of responses ($N$) affects different aggregation methods' performance on MATH with GPT 4o mini (tested on a 20% subset due to API constraints). Note that here we apply same diverse responses for all baselines to compare the aggregation methods, with self-consistency requiring one less model call as it uses majority voting for aggregation. We apply this setting for the rest of ablation studies to better compare the aggregation methods.

As shown in Figure 2, all methods improve significantly when moving from greedy decoding ($N = 1$) to multiple responses. Our method consistently outperforms the universal self-consistency baseline across different values of $N$, demonstrating the advantage of leveraging generative capabilities over attempting to select the best response. Both our method and self-consistency continue to benefit from more responses, indicating their ability to effectively aggregate information from a larger set of examples, while the universal self-consistency baseline shows diminishing returns and even a slight decline at larger $N$, suggesting that selecting among many samples can be challenging.

**Sampling temperatures.** We investigate how sampling temperature affects the performance of different methods on GSM8K and GPQA with Llama 3 8B. All methods show a clear pattern where performance improves as temperature increases from 0.2 to 1.0, then degrades at higher temperatures. With low temperature, the responses lack diversity, resulting in similar performance across methods.

The Oracle performance at this temperature also indicates limited diversity in the candidate pool. As temperature increases to 1.0, all methods benefit from increased response diversity, with our method achieve comparable performance to Self-consistency on GSM8K and better on GPQA, while consistently outperforming universal self-consistency. Further increasing temperature leads to performance degradation due to the decreased quality of sampled responses.

Table 3: Performance comparison of different sampling strategies on GSM8K for Gemma 2 9B with $N = 3$. **Temp.** uses temperature sampling with $T = 0.7$, **Prompt** uses different prompt templates, and **Multi.** generates responses in different languages.

| Method | Temp. | Prompt | Multi. |
|---|---|---|---|
| Best-of-N (Oracle) | 92.49 | 92.95 | 92.80 |
| Self-Consistency | 88.48 | 88.86 | **89.84** |
| Uni. Self-Consistency | 88.25 | 88.17 | 89.16 |
| Ours | **89.61** | **89.23** | 89.31 |

**Sampling strategies.** Beyond temperature-based sampling, we explore two additional strategies for generating diverse responses: (1) **Prompt template variation** uses different prompt templates to obtain diverse responses with distinct format. (2) **Multilingual generation** makes use of the multilingual capabilities of model, prompting them to answer the question in different languages.

Table 3 shows that all three sampling strategies achieve comparable performance on GSM8K using Gemma 2 9B with $N = 3$. Our method maintains strong performance across all strategies, indicating our method's robustness to the choice of sampling method. The multilingual sampling leads to slightly better oracle and self-consistency performance, possibly because responses in different languages provide more diversity. However, it require multilingual capabilities of the LLM.

Table 4: Performance comparison with adjusted model calls on GSM8K, MATH, and GPQA using GPT-4o Mini. GSA uses 5 model calls while Self-Consistency uses 6 calls, resulting in lower computational cost for GSA. All scores are accuracy percentages.

| Method | GSM8K | MATH | GPQA |
|---|---|---|---|
| Greedy | 93.48 | 76.54 | 41.07 |
| Self-Consistency (6 calls) | **94.84** | 77.85 | 40.85 |
| GSA (5 calls) | 94.77 | **78.33** | **42.17** |

**Adjusted model calls**. While our main experiments fix the number of model calls to ensure fair comparison across methods, we acknowledge that GSA incurs additional input tokens during the aggregation step. However, input token costs are typically much lower than output token costs (approximately 25% of output token pricing), making the overall cost difference modest.

Nevertheless, to better demonstrate GSA's effectiveness, we conducted additional experiments with adjusted model calls. Specifically, we tested GSA with 5 model calls against self-consistency with 6 model calls on GSM8K, MATH, and GPQA. Under this setting, GSA actually incurs lower computational costs than self-consistency while maintaining superior performance. As shown in Table 4, GSA consistently outperforms self-consistency on MATH and GPQA while remaining competitive on GSM8K.

## 4.4 Discussions

**Likelihood distribution.** To further investigate our hypothesis that LLMs are better suited for generating new responses than discriminative task, we analyze the normalized negative log-likelihood (NLL) distributions of responses produced by our method and the universal self-consistency on the Alpaca eval benchmark using Llama 3 8B. Figure 4 shows that our approach yields lower NLL values compared to universal self-consistency, indicating higher model confidence when generating new responses than when making selections.

We note that while lower NLL values may correlate with improved response quality, they alone are insufficient for optimal response selection. Prior work (Wang et al., 2023) has shown that simply selecting responses with the lowest NLL values performs substantially worse than self-consistency.

**Fine-grained Performance Analysis** To gain deeper insights into how our method and universal self-consistency utilize multiple candidate responses, we conduct a detailed comparative analysis on GPQA and MBPP with Llama 3 8B. We first categorize test samples based on the number of correct

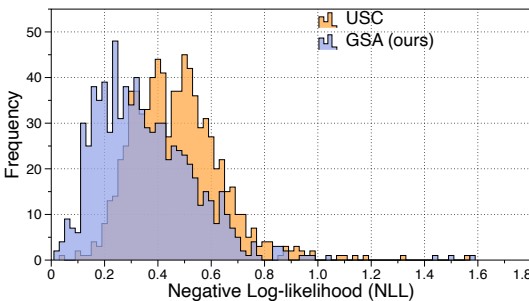

Figure 4: Distribution of normalized negative log-likelihood scores for responses generated on the Alpaca eval using Llama 3 8B. Lower NLL indicates higher model confidence in token generation.

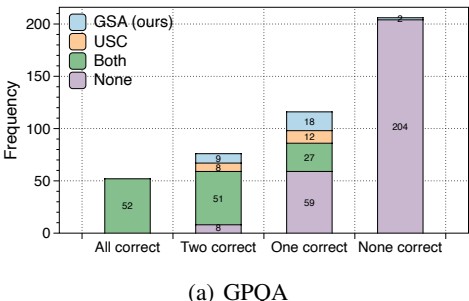

(a) GPQA

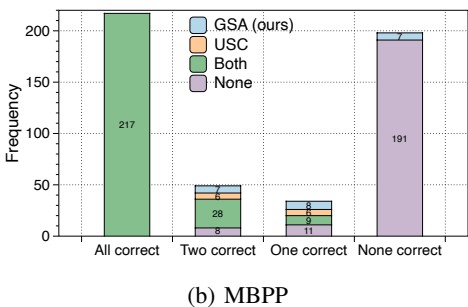

(b) MBPP

Figure 5: Comparison of our method vs. universal self-consistency on GPQA and MBPP with Llama 3 8B. Test cases are grouped by how many of the three candidates are correct. For each group, we show the number of samples where both methods, only ours, only universal self-consistency, or neither succeeded.

responses among the three candidates (3, 2, 1, or 0 correct). For each category, we then analyze four possible outcomes: both our method and universal self-consistency succeed, only our method succeeds, only universal self-consistency succeeds, or neither method succeeds.

Figure 5 visualizes this breakdown. When all three candidates are correct, both methods consistently produce correct answers. When two candidates are correct, both methods perform well, with our approach succeeding on slightly more samples than universal self-consistency. The advantage of our approach becomes more significant when only one candidate is correct. Notably, our method can solve some cases where none of the original candidates were correct, demonstrating its ability to synthesize a correct solution even from incorrect examples. In contrast, universal self-consistency is limited to selecting from existing candidates and hence cannot succeed when all candidates are incorrect.

## 5 CONCLUSION

In this paper, we introduced Generative Self-Aggregation (GSA), a novel prompting method that improves LLMs' performance without relying on discriminative judgments and can be applied to open-ended tasks. Our approach demonstrates that LLMs can effectively aggregate information from multiple solutions through generative processes, utilizing diverse reasoning paths to produce enhanced responses. GSA requires neither additional training nor external feedback, making it readily applicable across different model architectures and domains. Our extensive empirical evaluation across diverse tasks demonstrates that GSA outperforms existing self-correction and universal self-consistency methods that rely on LLMs' discriminative capabilities. As future work, GSA could be used to generate high-quality supervised data for model fine-tuning, and specialized training focused on enhancing LLMs' aggregation capabilities may further improve performance.

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

## A  APPENDIX

### A.1  CASE STUDY

We present two examples on coding and mathematical reasoning tasks to provide insights into how our method operates. As shown in Figure 6, we demonstrate how GSA effectively aggregates information from diverse candidate solutions to generate improved responses. In the coding task, the diverse candidates showcase different strategies, each with distinct limitations. Our method aggregates these approaches into an improved solution that combines simplicity of the first solution and the counting in the second. In the math task, while all candidates correctly determine that Jame's cousin's age in 8 years, two responses make errors in calculating the current age difference. The third response reaches the correct answer but employs a less intuitive algebraic approach using variables. Our method's response preserves the more straightforward calculation path seen in the first two responses, and correctly derive the final answer.

### A.2  IMPLEMENTATION DETAILS AND PARAMETERS

We conduct all experiments using inference-only settings, utilizing A6000 40GB GPUs and vLLM for efficient inference with open-source models, while accessing GPT-4o-mini through the OpenAI API service. For mathematical reasoning tasks, we set the maximum new token length to 2048, extending it to 4096 for GPQA and MMLU. When evaluating open-ended tasks (MT-bench, Alpaca eval, MBPP), we adhere to the default settings specified by each benchmark's evaluation convention. For all open-source models, we maintain a consistent top-p value of 0.95 during inference.

The temperature settings for candidate generation are tuned based on empirical performance of self-consistency and our method across different models and tasks. For LLaMA-3, we employ temperatures of 1.0 for GSM8K and GPQA, 0.7 for MATH, SVAMP, MMLU, and MT-bench, 0.5 for MTbench, and 0.8 for Alpaca eval. GPT-4o-mini uses a consistent temperature of 0.7 across all tasks except Alpaca eval, where we set it to 1.0. Gemma-2 maintains a uniform temperature of 0.7. Qwen-2.5 uses task-specific settings: 0.7 for GSM8K and MT-bench, 1.0 for GPQA, and 0.5 for MBPP. For the aggregation step, we employ greedy decoding (temperature = 0) for closed-ended tasks while maintaining task-specific temperature settings for open-ended benchmarks.

### A.3  LLM USAGE

In preparing this paper, large language models (LLMs) such as ChatGPT were used only for light editing purposes, including minor grammar checking and sentence polishing. No part of the research ideation, methodology design, experimental execution, or analysis was conducted with the assistance of LLMs.

### A.4  FULL SETS OF PROMPTS

We list the full details of the prompts used for candidates generation, universal self-consistency and our method on each task. We apply zero-shot setting for all baselines and hence do not require constructing any few-shot examples.

### A.5  GSM8K AND SVAMP

```
Question: {question}
Please put the final answer at the end of your response in the format "
Answer: <number>". Let's solve this step by step:
```

Listing 1: Candidates generation prompt for GSM8K and SVAMP

```
Question: {question}
Here are some potential responses:
{responses_text}
```

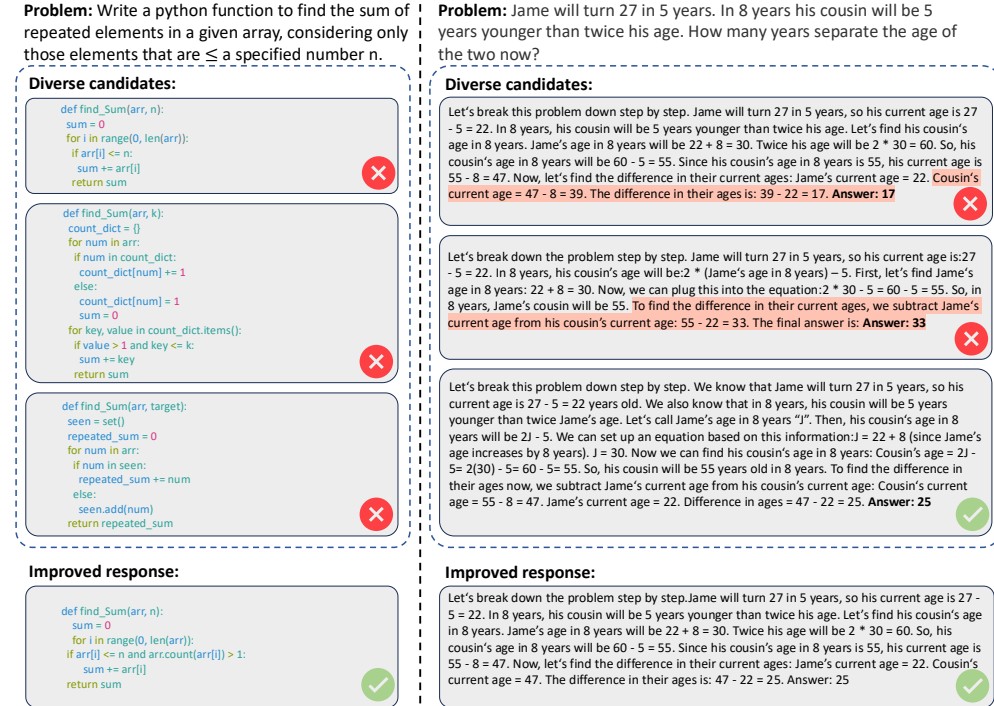

Figure 6: Examples of successful GSA applications. Left: Combining different coding approaches to create a simpler and correct implementation. Right: Preserving clear reasoning structure while avoiding calculation errors.

```
Given these solutions, please consider their consistency, and please
provide a correct solution to the question with clear reasoning and step-
by-step calculations.
Please put the final answer at the end of your response in the format "
Answer: <number>".
```

Listing 2: Aggregation prompt for GSM8K and SVAMP

```
Question: {question}
Here are some potential responses:
{responses_text}

Given these solutions, please consider their consistency and choose a
correct one. Give me clear explanation of your choice and put the index
of the correct answer at the end of the response. Please put the index in
 the format "Index: <index>". The index should be in the range of 1 to {
num_responses}.
```

Listing 3: Universal Self-Consistency prompt for GSM8K and SVAMP

For prompting variation in ablation study, we use the following prompt:

```
# Prompt 1
Question: {question}
Please put the final answer at the end of your response in the format "
Answer: <number>". Let's solve this step by step:

# Prompt 2
Question: {question}
Imagine you are explaining this problem to a student learning math for
the first time. Be clear and concise, and end your explanation with "
Answer: <number>".
```

```
# Prompt 3
Question: {question}
Solve the problem step by step, checking for potential errors along the
way. Provide the final answer at the end: "Answer: <number>".

# Multilingual
[{language}] Question: {question}
Please put the final answer at the end of your response in the format "
Answer: <number>". Let's solve this step by step using {language}:
```

Listing 4: Prompt variation for GSM8K

## A.6  MATH

```
Question: {question}
Please put the final answer at the end of your response in the form of \\
boxed{...}. Let's solve this step by step:
```

Listing 5: Candidates generation prompt for MATH

```
Question: {question}
Here are some potential responses:
{responses_text}

Given these solutions, please consider their consistency, and please
provide a correct solution to the question with clear reasoning and step-
by-step calculations.
Please put the final answer at the end of your response in the form of \\
boxed{...}.
```

Listing 6: Aggregation prompt for MATH

```
Question: {question}
Here are some potential responses:
{responses_text}

Given these solutions, please consider their consistency and choose a
correct one. Give me clear explanation of your choice and put the index
of the correct answer at the end of the response. Please put the index in
 the format "Index: <index>". The index should be in the range of 1 to {
num_responses}.
```

Listing 7: Universal Self-Consistency prompt for MATH

## A.7  GPQA

```
Question: {question}
Choices: {choices}

Please select an answer for the question from the above choices. Put the
final answer as a **single letter** at the end of the response in the
format "The correct answer is (insert answer here)". Let's think step by
step:
```

Listing 8: Candidates generation prompt for GPQA

```
{question and choices}

Here are some potential responses:
{responses_text}
```

```
Given these solutions, please analyze their consistency and correctness,
and then provide a correct solution with clear reasoning.
Put the final answer as a single letter at the end of your response in
the format "The correct answer is (insert answer here)".
```
Listing 9: Aggregation prompt for MMLU

```
{question and choices}

Here are some potential responses:
{responses_text}

Given these solutions, please consider their consistency and choose a
correct one. Give me clear explanation of your choice and put the index
(1-{num_responses}) of the correct answer at the end of the response.
Put the index of the correct answer as a single number in the format "The
 correct index is (insert index here)".
```
Listing 10: Universal Self-Consistency prompt for GPQA

## A.8 MMLU

```
Question: {question}
Choices: {choices}

Please select an answer for the question from the above choices. Put the
final answer as a **single letter** at the end of the response in the
format "The correct answer is (insert answer here)". Let's think step by
step:
```
Listing 11: Candidates generation prompt for MMLU

```
{question and choices}

Here are some potential responses:
{responses_text}

Please review the given solutions, and then provide a correct answer with
 clear reasoning.
Put the final answer as a single letter at the end of your response in
the format "The correct answer is (insert answer here)".
```
Listing 12: Aggregation prompt for MMLU

```
{question and choices}

Here are some potential responses:
{responses_text}

Please review the given solutions, and then give me the index (1-{
num_responses}) of the correct answer at the end of the response.
Put the index of the correct answer as a single number in the format "The
 correct index is (insert index here)".
```
Listing 13: Universal Self-Consistency prompt for MMLU

## A.9 MT-BENCH

```
{query}
Below are some responses to this instruction:
{responses_text}
```

```
Please review the above responses and generate a better response to the
instruction: <{query}>.
```

Listing 14: Aggregation prompt for MT-bench

```
{query}
Below are some responses to this instruction:
{responses_text}

Please review the above responses and choose a best response by providing
 the index (1-{n_responses}) of the best response. Please put the index
at the end of your response in the format "Index: <number>"."""
```

Listing 15: Universal Self-Consistency prompt for MT-bench

### A.10 ALPACA EVAL

```
###Instruction:
1) **Review** the following problem and the reference solutions provided.
2) **Provide** your own answer to the problem.
3) **Provide** a brief explanation of your reasoning.

###Reference Solutions:
{references_text}
###Input:
Here is the problem:
{question}
```

Listing 16: Aggregation prompt for Alpaca Eval

```
### Instruction:
1) **Review** the following problem and the reference solutions provided.
2) **Provide only** the index number of the best solution of the correct
solution in the format "Index: <number>".
3) **Provide** a brief explanation of your reasoning.

### Reference Solutions:
{solutions_text}
### Input:
Here is the problem:
{question}
```

Listing 17: Universal Self-Consistency prompt for Alpaca Eval

### A.11 MBPP

```
Here is the problem:
{prompt}
### Reference Solutions:
{references_text}
### Instructions:
1. Review the above solutions.
2. **Generate** a Python function that solves the Problem.
3. **Provide** a brief explanation of your reasoning.
4. **Ensure** your code is enclosed within a '''python''' code block.
```

Listing 18: Aggregation prompt for MBPP

```
### Instruction:
1) **Review** the following problem and the reference solutions provided.
2) **Provide only** the index number of the best solution of the correct
solution in the format "Index: <number>".
```

```
3) **Provide** a brief explanation of your reasoning.

### Reference Solutions:
{solutions_text}
### Input:
Here is the problem:
{prompt}
```

Listing 19: Universal Self-Consistency prompt for MT-bench

