# OpenReview forum: "LLMs Can Generate a Better Answer by Aggregating Their Own Responses"
_ICLR.cc/2026/Conference — ICLR 2026 Conference Withdrawn Submission_

### Official Review · Reviewer_by4h · 2025-10-29

**Soundness:** 2
**Presentation:** 2
**Contribution:** 1
**Rating:** 2
**Confidence:** 5

**Summary:**

While Large Language Models (LLMs) are highly capable, they often need help with complex problems. Popular methods like self-correction and selection fail when the LLM is tasked with evaluating its own work, due to a lack of training in discriminative judgment.
This paper introduces Generative Self-Aggregation (GSA), a new prompting technique that improves answers without needing the model to judge them. Instead, GSA:
1.Generates multiple diverse responses.
2.Aggregates them by synthesizing a new, improved answer based on all the generated text.
Unlike self-consistency, which relies on majority voting for specific answers, GSA is more flexible and works on open-ended tasks. Tests show it boosts performance in areas like math, knowledge questions, code, and dialogue.

**Strengths:**

1.Simplicity and efficiency: As a prompting method, GSA is simple to implement and does not require additional training, fine-tuning, or external models, making it highly efficient and accessible.
2.Generality and flexibility: Unlike Self-Consistency (SC), which is restricted to tasks with verifiable outputs (like multiple-choice), GSA's generative aggregation mechanism makes it applicable to a wide range of open-ended tasks, such as code synthesis and conversational responses.

**Weaknesses:**

1.Dependence on generation quality: The effectiveness of GSA is contingent on the quality and diversity of the initially sampled responses. If the model consistently generates flawed or homogeneous outputs for a given prompt, the aggregation process may be unable to synthesize a correct or improved solution (a "garbage-in, garbage-out" risk).

2.Lack of explicit error correction: While bypassing the need for discriminative judgment is a strength, it is also a limitation. The method does not explicitly identify or correct errors from the sampled responses; it relies on the generative process to implicitly overcome them, which may not be as reliable for specific, factual inaccuracies.

3.Limited contribution: The proposed method is straightforward and lacks sufficient innovative points.

**Questions:**

see the weakness

---

### Official Review · Reviewer_vDnS · 2025-10-31

**Soundness:** 3
**Presentation:** 2
**Contribution:** 1
**Rating:** 2
**Confidence:** 3

**Summary:**

This paper introduces a novel prompting method, Generative Self-Aggregation(GSA), which samples multiple diverse responses from LLM and then generates an improved answer based on the sampled responses.

**Strengths:**

- GSA only adds one aggregation prompt; no reward model, no verifier, no extra model. This makes it easy to drop into existing pipelines that already sample multiple candidates.
- Because it aggregates process rather than final tokens, GSA handles open-ended and code-generation tasks where voting is ill-defined; the paper shows gains on MT-Bench, AlpacaEval, and MBPP.

**Weaknesses:**

- Novelty is incremental. The core idea is a scope extension of self-consistency / universal SC to open-ended tasks, not a fundamentally new test-time reasoning paradigm. The paper even reuses the same candidate pool as SC.
- With 4 calls, GSA frequently ties or barely beats SC or even greedy: e.g., Llama-3-8B MMLU 65.62 vs SC 65.62; GPT-4o-mini MATH 78.25 vs SC 77.26; Llama-3-8B MATH 32.46 vs SC 31.68. These are real but slim deltas.
- The ablation (fix N=3, vary temperature) shows a trend, but the paper does not define or report an actual diversity metric.

**Questions:**

- Temperature ablation fixes N=3 and focuses on GSM8K/GPQA w/ Llama-3-8B. Why N=3? Do you observe interaction effects between N and temperature/diversity?
- Authors attribute GSA’s gains to “increased response diversity,” but there is no quantitative measure in the paper.

---

### Official Review · Reviewer_HWLD · 2025-11-01

**Soundness:** 3
**Presentation:** 2
**Contribution:** 2
**Rating:** 2
**Confidence:** 4

**Summary:**

This paper introduces Generative Self-Aggregation (GSA), a two-stage prompting strategy intended to improve an LLM’s response quality without asking the model to explicitly judge or rank candidates. Step (1) samples several diverse responses; step (2) feeds those responses back as context and asks the model to synthesize a new, improved response—leveraging generative next-token prediction rather than discriminative scoring. The authors position GSA as a generalization of self-consistency to open-ended tasks, and as an alternative to self-refine/choose-from-N methods that rely on an LLM’s judging ability. They evaluate across math (GSM8K, MATH, SVAMP), knowledge (MMLU, GPQA), and open-ended tasks (MT-Bench, AlpacaEval, MBPP) using Llama-3-8B, Gemma-2-9B, Qwen-2.5-14B, and GPT-4o-mini. GSA typically improves over greedy decoding, Self-Refine, and Universal Self-Consistency, matches or beats Self-Consistency where applicable, and extends to open-ended tasks where SC is not applicable.

**Strengths:**

S1. Simple idea with broad applicability: The paper’s core insight—aggregate by generation rather than select by judgment—is crisply articulated and easy to implement. The method is defined as “diverse response generation” followed by “context-enriched response synthesis,” with concrete prompt templates provided for each benchmark in the appendix, aiding reproducibility. Figure 1 and the worked examples make the intuition tangible (e.g., synthesizing from multiple solution paths to fix earlier missteps).

S2. Comprehensive Evaluation: The authors show their main results across four models and 7-8 diverse domains (math, knowledge, science and open-ended tasks). It is great that the authors also perform cost-aware comparison by fixing model calls (N=4) for fairness and further provides an “adjusted calls” comparison in Table 4. The authors perform meaningful ablations showcasing the effects of number of responses (Figure 2), temperature (Figure 3), and sampling strategies (template variation and multilingual sampling in Table 3) demonstrate robustness to these knobs. It is also useful detail that GSA can succeed even when all individual candidates are wrong, which selection-based methods cannot.

S3. Reproducibility aids: The appendix lists full prompts, implementation details (vLLM, temperatures, max tokens), and case studies. This level of detail is helpful for practitioners to try GSA quickly.

**Weaknesses:**

W1: My main concern with the paper is that it has very thin contribution. The improvement is minimal (< 1% in most cases compared to SC and also seen in Fig. 2 and Fig. 3). Further, numbers reported without error bars on smaller datasets (like GPQA) make the improvement of GSA even harder to discern. The authors say that SC's "application to open-ended tasks remains challenging due to the lack of clear voting mechanisms." While I acknowledge this, there are some indirect ways to go about this like embedding the responses and choosing the cluster with the highest count (or using a LLM-as-a-judge to perform semantic equivalence between two responses and then doing self-consistency).

W2: Apart from greedy decoding, evaluation at recommended temperature, top_p, etc. (for eg: Qwen recommends temp=0.7, top_p=0.8, etc.) should also be done and similarly added to Tables 1/2.

I also noticed some incorrect numbers. Following Qwen2.5 technical report (Table 7), the reported numbers in your paper seem to be lower (i.e., official report's normal evaluation has higher numbers than your proposed method):

| Benchmark | Official Report | Your Reporting (Greedy) |
|-----------|----------------:|---------------:|
| MBPP      |            82.0 |          72.00 |
| GPQA      |            45.5 |          39.51 |

This might be due to difference in sampling by the Qwen team. Thus, I would also recommend adding evaluation for each model in its recommend configuration (as suggested by their company). Please fix this.

W3: Benchmark sampling choices and statistical rigor: For MMLU, only 10% of the test set is evaluated; for MATH a 500-sample subset is used. There are no confidence intervals or statistical tests across runs/seeds. Please report mean +- std. err.

W4. Missing order-sensitivity: Aggregation may be sensitive to ordering of candidates. Was this factor studied? How sensitive is GSA to candidate ordering? I would request the authors to add more empirical analysis on candidate order like test best-candidate dropped vs. worst-candidate dropped or what fraction of the final response is composed from the each candidate (in their order).

W5: For Alpaca eval, why is GPT-4 used as judge for evaluating GPT-4o-mini (which came out long after GPT-4). GPT-4 is old now (by today's LLM standards) and may even be worse than Qwen2.5-14B on some tasks. At least GPT-4o should be used or similarly capable open-source model like DeepSeek-v3.1 if proprietary cost is an issue.

### Presentation:

1. In L228-230, what is wolves, cougars, etc.? The whole paragraph is unreadable.

2. Minor grammar: “generate an new response”, “we experiment our method” → “experiment with our method”, “number of class” → “calls”; a few curly quotes (Let‘s) and spacing inconsistencies.

3. Consistency: “MT-bench” vs “MTbench”; “Jame” vs “James” in the appendix example (if that’s not from the dataset verbatim).

**Questions:**

Q1: MBPP metric: Is MBPP reported as pass@1 using the official unit tests? If not, please detail the exact measurement.

---

### Official Review · Reviewer_Ge56 · 2025-11-05

**Soundness:** 2
**Presentation:** 2
**Contribution:** 1
**Rating:** 2
**Confidence:** 3

**Summary:**

The paper studies how LLMs can be used to better one-shot answer questions by processing their generated responses. Previous such techniques include self-consistency, self-correction and choose-from-N methods. They require either verifiable answers (e.g. math reasoning tasks) or discriminatively capable LLMs, which may be lacking. To address this, the paper proposes the GSA method which first generates multiple LLM responses, passes all of them in context to the LLM (along with the question) prompting it to generate a solution to the question.
The authors conduct experiments using 4 LLMs and 8 benchmark datasets showing performance improvements by GSA compared to previous techniques.

**Strengths:**

The paper proposes a simple response-aggregation based technique for improving an LLM’s answering capabilities, which can be effective and more generally applicable in comparison to previous methods.

**Weaknesses:**

1. The method is fairly straightforward and there seems to be little insight into why it supposedly works.
2. The empirical performance gains on the various benchmarks are not very substantial and seem inconsistent over model and datasets. In particular, the performance of GSA is similar to that of Self-consistency on the non-open ended datasets.
3. The error bars are not provided.
4. Some parts of the paper require more explanation (see questions to authors).

**Questions:**

1. Which model is used to predict the index as mentioned in line 271?
2. How is the Best-of-N oracle implemented for the open ended tasks?
3. While the experiments fix the number model calls, one ablation could be to also account for the larger context length of GSA. Have the authors considered this aspect?

---

### Note · Authors · 2025-11-12

I have read and agree with the venue's withdrawal policy on behalf of myself and my co-authors.